# Identification of RUVBL1 and RUVBL2 as Novel Cellular Interactors of the Ebola Virus Nucleoprotein

**DOI:** 10.3390/v11040372

**Published:** 2019-04-23

**Authors:** M. Jane Morwitzer, Sarah R. Tritsch, Lisa H. Cazares, Michael D. Ward, Jonathan E. Nuss, Sina Bavari, St Patrick Reid

**Affiliations:** 1Department of Pathology & Microbiology, University of Nebraska Medical Center, Omaha, NE 68198-5900, USA; melody.morwitzer@unmc.edu; 2United States Army Medical Research Institute of Infectious Diseases, Fort Detrick, Frederick, MD 21702-5011, USA; srtritsch@gmail.com (S.R.T.); lisa.h.cazares.ctr@mail.mil (L.H.C.); protid@comcast.net (M.D.W.); jonathan.nuss.iii@gmail.com (J.E.N.); sina.bavari.civ@mail.mil (S.B.)

**Keywords:** Ebola, NP, RUVBL1, RUVBL2, R2TP, AAA+ proteins

## Abstract

Ebola virus (EBOV) is a filovirus that has become a global public health threat in recent years. EBOV is the causative agent of a severe, often fatal hemorrhagic fever. A productive viral infection relies on the successful recruitment of host factors for various stages of the viral life cycle. To date, several investigations have discovered specific host-pathogen interactions for various EBOV proteins. However, relatively little is known about the EBOV nucleoprotein (NP) with regard to host interactions. In the present study, we aimed to elucidate NP-host protein-protein interactions (PPIs). Affinity purification-mass spectrometry (AP-MS) was used to identify candidate NP cellular interactors. Candidate interactors RUVBL1 and RUVBL2, partner proteins belonging to the AAA+ (ATPases Associated with various cellular Activities) superfamily, were confirmed to interact with NP in co-immunoprecipitation (co-IP) and immunofluorescence (IF) experiments. Functional studies using a minigenome system revealed that the siRNA-mediated knockdown of RUVBL1 but not RUVBL2 moderately decreased EBOV minigenome activity. Super resolution structured illumination microscopy (SIM) was used to identify an association between NP and components of the R2TP complex, which includes RUVBL1, RUVBL2, RPAP3, and PIH1D1, suggesting a potential role for the R2TP complex in capsid formation. Moreover, the siRNA-mediated knockdown of RPAP3 and subsequent downregulation of PIH1D1 was shown to have no effect on minigenome activity, further suggesting a role in capsid formation. Overall, we identify RUVBL1 and RUVBL2 as novel interactors of EBOV NP and for the first time report EBOV NP recruitment of the R2TP complex, which may provide novel targets for broad-acting anti-EBOV therapeutics.

## 1. Introduction

Ebola virus (EBOV) is in the family Filoviridae within the order Mononegavirales [1]. The virus is the causative agent of severe, often fatal, hemorrhagic fever in humans, with case fatality rates as high as 90% [2]. Since its emergence in 1976 [3], sporadic localized outbreaks have typically characterized EBOV, but the unprecedented spread of the 2013–2016 West African epidemic and its high death toll highlights the global public health threat that it poses [4]. The recent 2017 and 2018 outbreaks in Democratic Republic of the Congo further emphasize the urgency for EBOV therapeutic interventions [5,6,7]. Despite this, there remains no licensed therapeutics available, although a number of promising candidates are currently under investigation [8,9]. Novel therapeutic targets will thus be useful to the continued development of effective EBOV countermeasures. 

The negative-sense, single-stranded RNA (ssRNA) genome of EBOV is approximately 19 kb and encodes seven genes: Nucleoprotein (NP), virion protein 35 (VP35), VP40, glycoprotein (GP), VP30, VP24, and the RNA-dependent RNA polymerase (L) [1]. Owing to a relatively limited genome, host factors are essential for a productive viral infection. Notably, several investigations have discovered specific host-pathogen interactions in the EBOV lifecycle. At present, much is known about the host interaction partners of individual EBOV proteins including VP35, VP40, GP_1,2_, VP30, and VP24 [10,11,12,13,14,15,16]. This is highlighted by a recent study that showed interactome mapping of EBOV proteins and demonstrated that host protein RBBP6 inhibits EBOV RNA synthesis by competing with NP for binding to VP30 [17]. Besides the fundamental roles that NP plays in transcription and capsid assembly, relatively little is known about NP-host interactions [18,19,20,21]. A recent study, however, sought to define the cellular interactome of NP. Among the identified NP-host protein interactors, the chaperone protein Hsp70 was shown to bind and impact NP stability [22]. In addition, NP has been shown to recruit host PP2A-B56 phosphatase to modulate VP30 transcriptional activity [21]. Moreover, Staufen1 was shown to be required for efficient EBOV RNA synthesis via its interactions with multiple components of the replication complex, including NP [23]. Beyond these findings, NP-host protein-protein interactions (PPIs) remain unexplored. This is especially interesting given that, due to its gene order, NP is one of the most abundant proteins during the course of infection [1,24,25].

NP is the critical protein that drives the assembly of the nucleocapsid (NC) [18,26,27]. It interacts with the viral RNA genome, protecting it from degradation and recognition by the cellular innate immune factors [1,28,29]. NP, in the presence of VP24 and VP35, is sufficient to produce structures that are morphologically indistinguishable from the NCs formed during EBOV infection [18,19,26,27]. Furthermore, when NP is expressed by itself in bacteria or eukaryotic cells, it non-specifically binds to cellular RNAs, forming recombinant helical NP-RNA complexes that have identical morphology and stoichiometry to the authentic NP-viral RNA complexes [26,30,31,32,33]. The NP protein has a distinct localization pattern during infection or upon expression of the individual protein, forming distinct cytoplasmic inclusions [34,35,36]. 

In the current study, we aimed to further elucidate NP biology by identifying NP-host PPIs. We used affinity purification-mass spectrometry (AP-MS) to identify candidate NP cellular interactors, including RUVBL1 and RUVBL2. RUVBL1 and RUVBL2 interactions were verified by co-immunoprecipitation (co-IP) and immunofluorescence (IF) experiments. Furthermore, the utilization of an EBOV minigenome system showed that the siRNA-mediated knockdown of RUVBL1, but not RUVBL2, modestly decreased minigenome activity. In addition, the overexpression of RUVBL1 and RUVBL2 in the EBOV minigenome showed no effect on minigenome activity. Super resolution structured illumination microscopy (SIM) was then used to confirm the colocalization of NP with RPAP3 and PIH1D1, two other components of the RUVBL1/2-containing R2TP complex. The siRNA-mediated knockdown of RPAP3 and subsequent downregulation of PIH1D1 expression also had no effect on minigenome activity, instead suggesting a role in capsid assembly. Taken together, we identify RUVBL1/2 as novel interactors of EBOV NP and show recruitment of the R2TP complex, which may offer novel therapeutic targets for EBOV.

## 2. Materials and Methods

### 2.1. Cell Culture and Transfection

HeLa cells and HEK293T cells from the American Type Culture Collection (Manassas, VA, USA) were maintained in Dulbecco’s Modified Eagle Medium (DMEM) supplemented with 10% fetal bovine serum at 37 °C in a humidified 5% CO_2_ incubator. A conventional calcium phosphate transfection was used for initial AP-MS transfections. Subsequent experiments used a jetPRIME transfection reagent from Polyplus-transfection according to manufacturer’s recommendations (S.A., Illkirch, France).

### 2.2. Expression Vectors and Reagents

pCAGGS-HA-EBOV-NP was purchased from BEI (NR-49343). pCAGGS-NP-V5, pCAGGS-V5-VP30, and pcDNA3 vectors were generated at the USAMRIID. pCAGGS-FLAG-VP35 was a kind gift from Dr. Christopher F. Basler (Georgia State University). pCDNA-3xFLAG-Pontin and pCDNA-3xHA-Reptin were gifts from Dr. Steven E. Artandi (Addgene plasmid # 51635; # 51636) [37]. pCAGGS_L_EBOV and pCAGGS_3E5E_luciferase were gifts from Dr. Elke Mühlberger (Addgene plasmid # 103052; # 103055) [38]. The pRL-TK plasmid was kindly provided by Dr. Tsung-Hsien Chang (Kaohsiung, Taiwan) [39]. 

Mouse anti-HA (H3663), rabbit anti-HA (H6908), mouse anti-FLAG (F3165), rabbit anti-FLAG (F7425), mouse anti-V5 (V8012), rabbit anti-V5 (V8137), mouse anti-RPAP3 (WH0079657M1), rabbit anti-GAPDH (SAB2108668), and mouse anti-Calnexin (C7617) antisera were purchased from Sigma-Aldrich (ST. Louis, MO, USA). Rabbit anti-PIH1D1 (ab238862) antisera was purchased from Abcam (Cambridge, MA, USA). Rabbit anti-RUVBL1 (NBP2-20245) and rabbit anti-RUVBL2 (NBP1-40354) antisera were purchased from Novus Biologicals (Centennial, CO, USA). Goat anti-HA (A00168) antisera was purchased from GenScript (Piscataway, NJ, USA). Goat anti-rabbit HRP (65-6120), goat anti-mouse HRP (32430), goat anti-mouse Alexa Fluor 488 (A32723), goat anti-rabbit Alexa Fluor 568 (A11011), donkey anti-goat Alexa Fluor 647 (A21447), and Hoechst 33342 (H3570) were purchased from ThermoFisher Scientific (Grand Island, NY, USA). 

Silencer Select Negative Control No. 1 siRNA (4390843), Silencer Validated siRNA RUVBL1 (AM1331), Silencer Pre-designed siRNA RUVBL2 (AM16704), and Silencer Pre-designed siRNA RPAP3 (AM16708) were purchased from ThermoFisher Scientific. The Dual-Glo Luciferase Assay System (E2920) was purchased from Promega (Madison, WI, USA) and used according to manufacturer’s recommendations.

### 2.3. Gel-Liquid Chromatography/Tandem Mass Spectrometry Analysis

Affinity purified material and the empty vector control were resolved by sodium dodecyl sulfate polyacrylamide gel electrophoresis (SDS-PAGE) on 10–20% gradient polyacrylamide gels (ThermoFisher Scientific). Protein was visualized by Imperial Coomassie blue staining (ThermoFisher Scientific), and equal sized sections from the affinity purified sample were further analyzed by liquid chromatography/mass spectrometry (LC/MS/MS). Equal sized gel pieces were excised from this immune-precipitated sample and destained, dehydrated, and reductively alkylated by incubating samples in 2.5 mM DTT for 1 h at 60 °C and adding iodoacetamide to 10 mM for 1 h incubation in the dark. Samples were in-gel digested with trypsin during overnight incubations at 37 °C and individual digestates analyzed directly by LC/MS/MS. All LC/MS/MS experiments were conducted on an AmaZon ion trap mass spectrometer from Bruker Daltonics (Billerica, MA, USA) and interfaced with a nanoflow HPLC system (Ultimate 3000 nano, Dionex; Sunnyvale, CA, USA). Tryptic peptides were separated under nanoflow conditions (400 nL/min) using a trap loading column (Acclaim PepMap 100, 5 μm particle size and 300 Å pore size, Dionex) and resolved with a linear acetonitrile gradient (60 min, 0–45% acetonitrile, 0.1% formic acid) on a monolithic c18 reverse phase column (Acclaim PepMap100, 3 µm particle size, 100 Å pore size, Dioxnex). Tandem MS/MS data was acquired using data dependent acquisition by fragmenting the 4 most abundant ions in each spectral scan by collision induced dissociation (after 3 consecutive acquisitions, ions were excluded from further CID). Raw data was deconvoluted by an automated charge state and mass assignments using Bruker Daltonics software (DataAnalysis), and the MASCOT algorithm was used to search the SwissProt database (species: Homo Sapiens). Statistical cutoffs (*p*-values less than 0.01) were used to identify unknown host proteins.

### 2.4. Protein Co-Immunoprecipitation

HeLa cells (1 × 10^6^ cells) were transfected with the indicated plasmids using 3 µL of the transfection reagent jetPRIME (Polyplus) per 1 µg DNA per manufacturer’s instructions. The total amount of DNA for each transfection was kept constant in each experiment by complementing with empty vector. Twenty-four hours post-transfection, cells were lysed in a modified RIPA buffer (50 mM Tris-HCl pH 7.4, 150 mM NaCl, 1mM EDTA, 1% NP-40, 0.25% Na-deoxycholate) containing protease and a phosphatase inhibitor cocktail (Thermo Scientific, Waltham, MA, USA). A portion of ~10% whole cell lysate was reserved, and immunoprecipitation was performed using EZview Red ANTI-FLAG M2 Affinity Gel (Sigma), EZview Red Anti-HA Affinity Gel (Sigma), or Anti-V5 Agarose Affinity Gel (Sigma) according to manufacturer’s recommendations. After 1 h of bead incubation at 4 °C, the beads were left untreated or treated with 20 mg/mL of RNase A (Qiagen) and incubated at 37 °C for 30 min. Beads were washed 4 to 5 times with TBS, re-suspended in 2X Lane Marker Reducing Sample Buffer (Thermo Scientific), and then subjected to SDS-PAGE and immunoblotting.

### 2.5. Immunoblotting

Cell lysates were subjected to SDS-PAGE and proteins were transferred onto PVDF membranes. The following primary antibodies were used for detection of proteins by immunoblot: Rabbit anti-HA, rabbit anti-FLAG, rabbit or mouse anti-V5, rabbit anti-RUVBL1, rabbit anti-RUVBL2, mouse anti-RPAP3, rabbit anti-PIH1D1, rabbit anti-GAPDH, and mouse anti-Calnexin. Blots were probed with primary antibodies either 1–2 h at room temperature or overnight at 4 °C. Secondary incubations were performed for 1–2 h at room temp using either goat anti-rabbit HRP or goat anti-mouse HRP. Radiance chemiluminescence substrate (Azure Biosystems; Dublin, CA, USA) was used to visualize protein on an Azure c600 imaging system.

### 2.6. Confocal Microscopy

HeLa cells (1 × 10^5^ cells) were grown on Fisherbrand Microscope Coverglass slides (Fisher Scientific) and transfected with the indicated plasmids using jetPRIME, according to manufacturer’s instructions. Vector controls were included for endogenous samples in the absence of HA-NP. Twenty-four hours post-transfection, cells were fixed in 4% PFA, permeabilized in 0.01% Triton X-100 (Fisher Bioreagents), and blocked in 3% bovine serum albumin (Fisher Bioreagents). The primary antibodies mouse anti-HA or goat anti-HA, rabbit anti-FLAG, rabbit anti-V5, rabbit anti-RUVBL1, rabbit anti-RUVBL2, mouse anti-RPAP3, and rabbit anti-PIH1D1 were used along with the secondary antibodies goat anti-mouse Alexa Fluor 488, goat anti-rabbit Alexa Fluor 568, and donkey anti-goat Alexa Fluor 647. Slides were probed with primary antibodies either for 1–2 h at room temperature or overnight at 4 °C. Secondary incubations were performed for 1–2 h at room temperature. Hoechst 33342 was used for a nuclear stain. Slides were mounted using Fluoromount Aqueous Mounting Medium (Sigma). Colocalization images were collected by a Zeiss LSM800 confocal microscope (Carl Zeiss MicroImaging). SIM images were collected with a Zeiss ELYRA PS.1 illumination system (Carl Zeiss MicroImaging), and SIM processing was performed with the SIM module of Zen BLACK software (Carl Zeiss MicroImaging).

### 2.7. siRNA Transfection and Minigenome Assay

HeLa cells (2.5 × 10^5^) were seeded in 12-well plates and transfected in suspension with 30 nM of siRNA against RUVBL1, 50 nM of siRNA against RUVBL2, a combination of siRNA against RUVBL1 and RUVBL2, or 80 nM of Silencer Select Negative Control No. 1 siRNA. For RPAP3 knockdown experiments, 30 nM of siRNA against RPAP3 was used, along with a siRNA control of 30 nM of Silencer Select Negative Control No. 1. The siRNA concentrations used were determined by previous optimization experiments. The minigenome assay was performed 24 h post-siRNA transfection using a RNA polymerase II-driven EBOV minigenome as previously described [38]. Briefly, HeLa cells were transfected with 125 ng pCAGGS-HA-NP, 125 ng pCAGGS-FLAG-VP35, 50 ng pCAGGS-V5-VP30, 50 ng pRL-TK, 500 ng pCAGGS-L, and 750 ng of pCAGGS-3E5E-luciferase. For the no L control, total DNA levels were kept constant by complementing transfections with empty-vector pcDNA3. For RUVBL1/2 overexpression in the minigenome, cells were left untransfected, or transfected with vector control (VC), or increasing amounts of FLAG-RUVBL1 (125, 250, and 500 ng) or HA-RUVBL2 (125, 250, and 500 ng). Twenty-four h after exogenous transfection, the minigenome components were transfected. Reporter activity was measured 48 hours post-transfection using the Dual-Glo Luciferase Assay System and a Tecan Spark microplate luminometer (Tecan Trading AG, Switzerland). Whole cell lysate was reserved from a representative replicate and subjected to immunoblotting as described above. To account for potential differences in transfection efficiency, firefly luciferase activity was normalized to *Renilla* luciferase values and plotted as fold activity calculated relative to the no L control. Standard error of the mean (SEM) values and paired, two-tailed t tests were calculated using GraphPad Prism 7.05 software (GraphPad Software, Inc.; San Diego, CA, USA). 

## 3. Results

### 3.1. Identification of NP Candidate Cellular Interactors by Affinity Purification-Mass Spectrometry (AP-MS)

To gain a more complete understanding of EBOV NP-host PPIs, we employed an AP-MS approach to identify candidate cellular interactors of NP. The affinity-tagged NP protein was over-expressed in HEK 293T cells, and interacting host proteins were identified following affinity purification (AP); empty vector, transfected control cells were also processed by identical AP procedures. The control and EBOV NP samples were then visualized by SDS-PAGE (Figure 1A). To identify candidates, the entire gel was excised, and the pieces were in-gel digested with trypsin. Individual digestates were then processed by nanoflow LC/MS/MS to identify the co-immunoprecipitated host proteins in each gel slice. The host cell proteins that were identified by this analysis are listed in Figure 1C. The connectivity of these proteins was assessed using known and predicted interactions in the STRING PPI database and constructed a PPI network (Figure 1B). Comparing these results to two previous studies that investigated NP cellular interactors revealed an overlap of seven cellular proteins that have a higher probability of forming protein-protein interactions, including TROVE2, TUBA1B, TUBB2C, IGF2BP1, PDHB, RUVBL2, and RUVBL1 [17,22]. A full list of identified interactors is included in Appendix A.

### 3.2. Validation of NP-Candidate Interactions by Co-Immunoprecipitation (Co-IP)

To validate candidate interactors of NP, we performed co-IPs. The NP candidate interactors selected for further investigation were RUVBL1 and RUVBL2. RUVBL2 was recently reported as a potential but unconfirmed NP interactor in a MS study [22]. RUVBL1/2 also play roles during Human adenovirus type 5 (HAdV-5), influenza A virus (IAV), human immunodeficiency virus type 1 (HIV-1), and hepatitis virus b (HBV) infections [40,41,42,43]. Furthermore, the two closely related proteins are co-expressed and share similar functional properties. RUVBL1/2 commonly share binding partners and play roles in various multiprotein complexes that are involved in diverse processes such as transcriptional regulation, chromatin remodeling, and ribonucleoprotein complex biogenesis [44,45]. As a result, we further investigated the NP-RUVBL1/2 association. NP is an RNA-binding protein, as are RUVBL1 and RUVBL2 [26,44,46]; therefore, after immunoprecipitation, samples were treated with or without RNase before immunoblot analysis to determine if the interactions were mediated by RNA binding (Figure 2A). HeLa cells were transfected with HA-NP and FLAG-RUVBL1 or NP-V5 and HA-RUVBL2. After 24 h, cellular proteins were extracted and applied to IPs using epitope-tagged beads. After the IPs and subsequent −/+ RNase incubation, whole cell lysates (WCLs) and IP samples were subjected to SDS-PAGE, and products were analyzed by immunoblot. FLAG-RUVBL1 was shown to be pulled down only in the presence of HA-NP, demonstrating specificity of binding, with WCLs confirming each protein’s expression (Figure 2B). Moreover, the interaction was kept intact after RNase treatment, indicating that NP-RUVBL1 interaction is not mediated by RNA (Figure 2B). Reciprocally, the FLAG IP demonstrated that HA-NP interacts with FLAG-RUVBL1 (Figure 2C). Similarly, NP-V5 was also shown to interact with HA-RUVBL2 (Figure 2D,E). These data confirm that NP interacts with RUVBL1 and RUVBL2 in an RNA-independent manner. 

### 3.3. Assessing Localization of RUVBL1 and RUVBL2 in the Presence of NP

To validate further the mass spectrometry and IP results, we used confocal microscopy to examine the localization of endogenous RUVBL1 and RUVBL2 in the absence or presence of HA-NP. HeLa cells were transfected with vector control or HA-NP. In the absence of HA-NP, both RUVBL1 and RUVBL2 showed strong nuclear localization as has been previously shown [47,48] (Figure 3A,B, top panels). Notably, the presence of HA-NP resulted in distinct re-localization of RUVBL1 to NP inclusions (Figure 3A, middle panels). Likewise, the colocalization of RUVBL2 and HA-NP was observed, although the relocalization of RUVBL2 was more peripheral to NP inclusions than RUVBL1 (Figure 3B, middle panels). To further demonstrate that RUVBL1/2 specifically bind to NP and not the HA tag, we included an additional control of another HA-tagged protein and co-stained for RUVBL1 or RUVBL2. Specifically, EBOV HA-VP35 localization was observed to be characteristically diffuse, with no colocalization between HA-VP35 and RUVBL1 or RUVBL2 (Figure 3A,B, bottom panels). That expected cellular localization of RUVBL1 and RUVBL2, with no indication of non-specific interaction, was observed when cells were transfected with vector control as well as HA-VP35, all of which further supports the specificity of RUVBL1/2-NP association. Taken together, these data strongly show NP association with RUVBL1/2.

### 3.4. Evaluating the Effects of RUVBL1 and RUVBL2 on EBOV Minigenome Activity

Since RUVBL1 and RUVBL2 are part of complexes that are involved in transcriptional regulation, chromatin remodeling, and RNA modification, we next assessed how the RUVBL1 and RUVBL2 interaction affects NP function by using an EBOV minigenome system (Figure 4A). In this minigenome system, the components required for EBOV RNA synthesis (NP, L, VP35, and VP30) are provided in trans with a miniaturized version of the EBOV genome, which is comprised of a firefly luciferase reporter gene flanked by the necessary EBOV genomic sequences for viral polymerase recruitment [38]. Notably, the EBOV minigenome of this particular system is under the control of RNA polymerase II. This recently developed system is an attractive alternative to previous T7 polymerase-driven EBOV minigenome systems because it utilizes endogenous polymerase and is in turn amenable to a wider array of cell lines [38]. The minigenome components are co-transfected into HeLa cells, and viral transcription is measured by firefly luciferase activity. To assess an effect on the minigenome by RUVBL1 or RUVBL2, an siRNA-mediated knockdown was performed 24 h before minigenome transfection. Forty-eight hours post minigenome transfection, a dual-luciferase assay was used to measure reporter activity. Notably, while the siRNAs were specific for the respective mRNAs, a simultaneous decrease in both RUVBL1 and RUVBL2 protein levels were observed regardless of which siRNA was used, as has been previously reported [37,49,50,51] (Figure 4B). The knockdown of RUVBL1 moderately decreased MG activity, whereas the knockdown of RUVBL2 did not have an effect on minigenome activity. Further, the combinatorial knockdown of RUVBL1 and RUVBL2 failed to decrease the minigenome activity. Though RUVBL1 and RUVBL2 generally act in complex, they have been shown to act independently [52,53,54]. That only RUVBL1 knockdown is observed to affect minigenome activity suggests RUVBL1 may drive the interaction with NP. This is consistent with the distinct colocalization observed for RUVBL1 in the presence of HA-NP as compared with the more peripheral colocalization of RUVBL2 with HA-NP inclusions (Figure 3). To further investigate the effect of RUVBL1 and RUVBL2 on minigenome activity, increasing concentrations of FLAG-RUVBL1 or HA-RUVBL2 were overexpressed 24 h before minigenome transfection. Interestingly, neither FLAG-RUVBL1 nor HA-RUVBL2 overexpression was observed to affect minigenome activity (Figure 4C). As such, while a statistically significant decrease in minigenome activity was observed upon the RUVBL1 knockdown, neither RUVBL1 nor RUVBL2 appear to be functionally relevant to EBOV transcription/replication. Altogether, these results suggest that RUVBL1/2 do not play a major role in EBOV transcription/replication. 

### 3.5. Identification of the R2TP Complex Recruitment to NP Inclusions

To further assess RUVBL1/2 in NP biology, we next sought to identify the multiprotein complex that RUVBL1/2 are part of when associating with NP. Besides a role in transcription, NP is the major capsid protein [18,26,27]. Since minigenome activity was modestly affected by the RUVBL1 knockdown, and the overexpression of RUVBL1/2 did not affect minigenome activity, we next turned our attention to capsid assembly. In particular, we looked for the involvement of RUVBL1/2 in complexes that are associated with macromolecular assembly because capsid formation is a multi-subunit assembly process. As proteins that belong to the AAA+ (ATPases Associated with various cellular Activities) superfamily, RUVBL1 and RUVBL2 could act in a number of ways with various complexes to influence NP biology [37,44,45,49,50,53,55]. The R2TP complex, comprised of RUVBL1, RUVBL2, RPAP3, and PIH1D1, became of interest because of its specialized co-chaperone activity to heat-shock protein 90 (Hsp90), which is associated with assembly and maturation of multi-subunit complexes and whose inhibition has been shown to reduce EBOV replication by an as yet unidentified mechanism [50,56,57,58,59,60,61,62,63]. Furthermore, the R2TP complex forms another more specialized co-chaperone complex, R2TP/prefoldin-like (PFDL). Interestingly, a recent study identified interactors of the R2TP/PFDL complex, including the chaperonin CCT/TRiC [63], which has recently been shown to control reovirus replication through outer-capsid folding [64]. Moreover, it is notable that TRiC/CCT components were present in the mass spectrometry dataset of candidate interactors (Figure 1B; Appendix A). To confirm an association of NP with the R2TP complex, we used SIM to assess NP localization with RPAP3 and PIH1D1. Both RPAP3 and PIH1D1 were observed to accumulate at NP inclusions, exhibiting very strong colocalization (Figure 5C). We also included vector controls with endogenous PIH1D1 and RPAP3 staining to evaluate any non-specific interactions, which were not observed (Figure 5A,B). Finally, we tested the effect of the RPAP3 knockdown on minigenome activity. It should be noted that the RPAP3 knockdown has previously been shown to destabilize PIH1D1, thus also causing decreased PIH1D1 levels [65]. The siRNA-mediated knockdown of RPAP3 and subsequent downregulation of PIH1D1 protein levels was found to have no effect on minigenome activity, indicating that these proteins have no function in EBOV transcription/replication (Figure 5D). Taken together with the confirmed interaction and colocalization of NP with RUVBL1/2, SIM clearly indicates an association between NP and R2TP complex, with minigenome data further supporting a role for capsid assembly rather than EBOV transcription/replication. This is the first report of NP recruitment of the R2TP complex and thus presents potential novel therapeutic targets for EBOV.

## 4. Discussion

EBOV infection often results in fatal illness in humans, and while there are promising vaccine and therapeutic candidates, there remains a need to develop effective EBOV therapeutics [8,9]. As obligate intracellular parasites, viruses rely on various host factors for productive viral infection, making host-targeted strategies an attractive therapeutic approach. Relatively little is known regarding host factors that interact with and influence EBOV NP functions. Identifying NP-host interactions will improve upon the current understanding of NP functions in EBOV biology, thus providing potential cellular targets for novel anti-EBOV therapeutics that target host proteins crucial to the EBOV life cycle [22,66,67]. In the present study, we used an AP-MS approach to identify candidate cellular interactors of NP. In particular, coIPs were used to confirm novel interactions between NP and RUVBL1 as well as RUVBL2, with strong colocalization between NP and both RUVBL1 and RUVBL2 also observed via IF. We posited that RUVBL1 and RUVBL2 play a role in EBOV transcription/replication, but we only observed a moderate decrease in minigenome activity when RUVBL1, but not RUVBL2, was knocked down. Further, the overexpression of RUVBL1 and RUVBL2 showed no effect on minigenome activity. Subsequently, SIM was used to show an association of NP inclusions with RPAP3 and PIH1D1, two other components of the RUVBL1/2-containing R2TP complex. The knockdown of RPAP3, which is also known to decrease PIH1D1 protein levels, also did not affect minigenome activity appreciably [65]. Since the R2TP complex is responsible for stabilization and maturation of macromolecular complexes, this suggests that the R2TP complex plays a role in EBOV capsid formation [50,56,57,58,59,60,61].

NP function is distinct in the replication cycle as it is a key component of the viral ribonucleoprotein complex, playing critical roles in protecting the RNA genome from degradation and cellular immune recognition, as well as mediating capsid assembly [1,18,26,27,28]. Any ancillary roles of NP remain to be determined. Filovirus RNA synthesis occurs in cytoplasmic inclusions where PPIs among replication complex constituents NP, VP35, VP30, and L are well characterized [68,69,70]. Notably, expression of VP35, VP30, and L proteins individually results in diffuse localization patterns of each, whereas the individual expression of NP results in the formation of cytoplasmic inclusions that resemble the sites of replication during infection. Moreover, NP co-expression with VP35, VP30, and L results in the co-localization of these proteins to the inclusions [25,71]. We identified an association of NP inclusions with the R2TP complex that suggests recruitment of this specialized Hsp90 co-chaperone complex by NP [62,72]. This is interesting given that NP appears to drive assembly of replication complexes and that the R2TP complex is known to be essential in the assembly of several macromolecular complexes [73]. Furthermore, Hsp90 has been shown to be important to EBOV biology; however, the mechanism has not been elucidated [62]. Though it has been suggested that Hsp90 stabilizes L, as has been demonstrated for vesicular stomatitis virus (VSV), it is also possible that the NP recruitment of its co-chaperone R2TP complex underlies the mechanism of Hsp90 function [74].

RUVBL1/2 have individually as well as dually been implicated in viral infections. A novel interaction between RUVBL1 and Human adenovirus type 5 (HAdV-5) E1A was recently identified to be important for suppression of interferon response [42]. RUVBL2 was shown to suppress influenza A virus (IAV) replication by interfering with NP oligomerization [75]. In contrast, RUVBL2 inhibitory activity on human immunodeficiency virus type 1 (HIV-1) Gag protein expression was shown to be important for efficient production of infectious progeny by balancing relative expression levels of the Gag and envelope protein (Env) [41]. A genome-wide RNAi screen identified RUVBL1 as a host factor that inhibits arbovirus infection in adult flies, mosquito cells, and mammalian cells [76]. Similarly, another genome-wide RNAi screen identified the chromatin-remodeling complex TIP60, which contains RUVBL1/2 proteins, as an inhibitor hepatitis virus b (HBV) transcription [43]. Viruses manipulate various host factors and cellular pathways to enable replication and production of infectious progeny; however, diverse viruses use differing mechanisms to co-opt host resources. Our findings suggest a beneficial role of RUVBL1/2 in EBOV biology. 

The identification of novel interactions between NP and RUVBL1/2 and the association observed between NP and the R2TP complex suggest potential novel therapeutic targets of EBOV. RUVBL1 and RUVBL2 are related AAA+ proteins that often exist in complex with each other. In fact, their ATPase activity has been shown to require the catalytic activity of both in complex, with ATP hydrolysis synergistically increasing upon hetero-oligomerization compared to that of the constituent proteins [77,78]. Because the RUVBL proteins are necessary for cell viability, the ability to interrogate the cellular contributions of RUVBL proteins has been limited for decades. A potent and selective inhibitor of the RUVBL1/2 complex, however, was reported this year, offering a therapeutic approach [79]. Moreover, the PIH1D1 component of the R2TP complex may offer a potential target, as the substrate specificity of the R2TP co-chaperone complex has been shown to be dependent on PIH1D1 interactions. Specifically, the N-terminal region of PIH1D1 contains a unique phosphopeptide binding domain that preferentially binds to acidic phosphorylated proteins. Thus, the phosphopeptide binding domain of PIH1D1 can conceivably be targeted by a small molecule mimetic to interfere with PPIs between the R2TP co-chaperone complex and substrates [80]. Of note, the C-terminal domain of NP is highly acidic and possesses multiple phosphorylation sites, which could account for the ability of NP to recruit the R2TP complex [81,82].

Comparing interactor candidates with two previous studies that used similar AP-MS approaches revealed an overlap of seven interactions including TROVE2, TUBA1B, TUBB2C, IGF2BP1, PDHB, RUVBL2, and RUVBL1 [17,22]. However, we confirmed novel interactions between NP and RUVBL1/2 and showed an association between NP and the R2TP complex for the first time. A further investigation of the other overlapping interactors may yield more insight into NP biology. Because data was not collected from cells infected with EBOV, functionally significant interactions that are mediated by infection/the presence of multiple viral proteins may have been missed. Moreover, any functional role of RUVBL1 in EBOV transcription/replication as suggested by the modest decrease in minigenome activity upon knockdown remains to be confirmed in the context of EBOV infection. Future studies must confirm relevance during EBOV infection as well as address the precise role of R2TP complex to EBOV biology, with a structural dissection of the interactions between NP and R2TP components to identify specific therapeutic targets.

In recent years, more cellular interactors of EBOV NP have been identified [21,22,23]. Though these studies have provided more insight into NP biology, much remains to be understood. In particular, the host factors and mechanisms co-opted by NP for replication and capsid assembly. Here, our data indicate that RUVBL1 and RUVBL2 interact with EBOV NP, the RUVBL1 knockdown moderately decreases EBOV transcription, and that NP associates with the RUVBL1- and RUVBL2-containing R2TP complex. While further functional analysis is necessary to assess the significance of R2TP members in EBOV biology, the identification of these novel NP cellular interactors, along with NP association with the R2TP complex, provides new potential targets for the development of host-based anti-EBOV therapeutics. Future studies will assay capsid assembly and incorporate viral infections to address the functional significance of R2TP members in NP biology. 

## Figures and Tables

**Figure 1 viruses-11-00372-f001:**
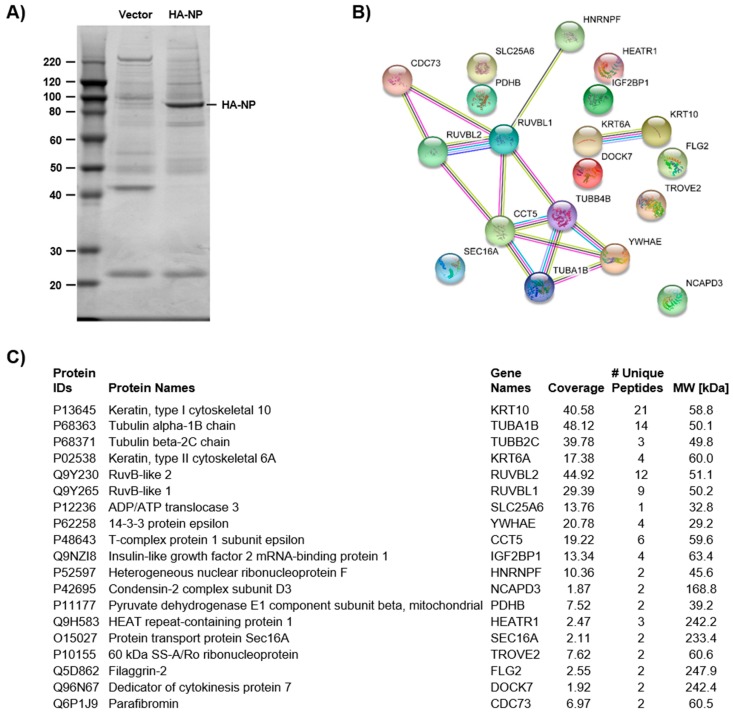
Identification of candidate Ebola virus (EBOV) nucleoprotein (NP)-interacting host proteins. (**A**) Silver stain of HA-NP immunoprecipitations. (**B**) STRING analysis of NP interactor candidates. (**C**) Candidate proteins identified by mass spectrometry analysis.

**Figure 2 viruses-11-00372-f002:**
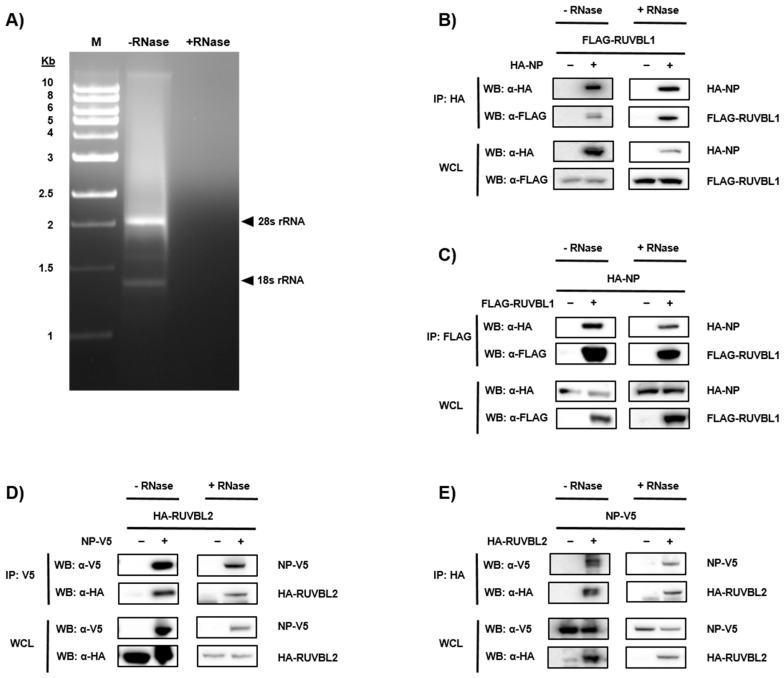
Validation of protein-protein interactions between EBOV NP and RUVBL1 and RUVBL2. (**A**) RNase treatment was confirmed using agarose gel electrophoresis on samples that were left untreated or treated with RNase (designated by − RNase or +RNase). (**B**) Confirmation of HA-NP and FLAG-RUVBL1 interaction in the absence or presence of RNase. Whole cell lysates (WCL) samples were run in parallel with immunoprecipitations (IP) samples to confirm appropriate protein expression. (**C**) Reciprocal validation of protein-protein interaction between HA-NP and FLAG-RUVBL1 by FLAG IP and −/+ RNase treatments. (**D**) Confirmation of NP-V5 and HA-RUVBL2 interaction by V5 IP and −/+ RNase treatments. (**E**) Reciprocal validation of protein-protein interaction between NP-V5 and HA-RUVBL2 by HA IP and −/+ RNase treatments. Monoclonal antibodies against HA, FLAG, and V5 were used to detect target proteins.

**Figure 3 viruses-11-00372-f003:**
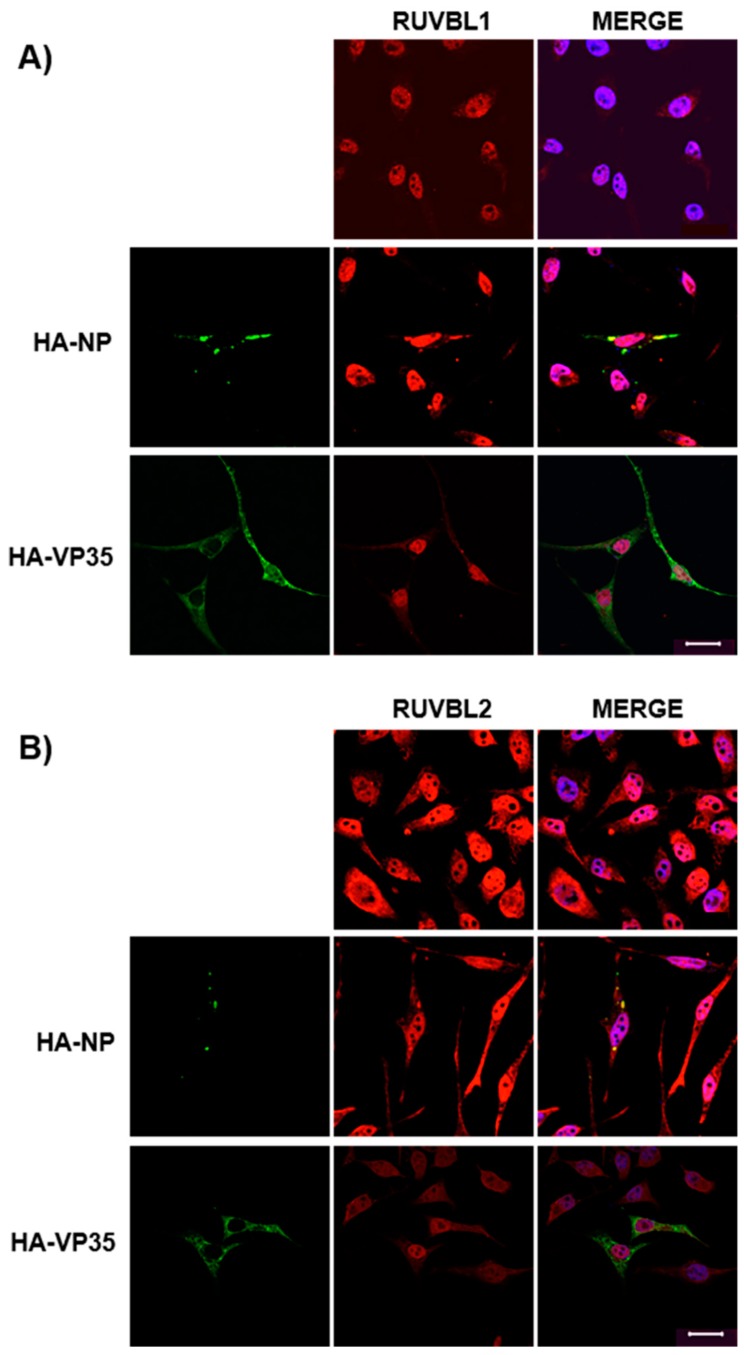
Endogenous RUVBL1 and RUVBL2 colocalize with HA-NP. HeLa cells were transfected with vector control, HA-NP, or HA-VP35. Twenty-four h later, the cells were fixed and processed for immunofluorescence detection of endogenous RUVBL1 or RUVBL2 in the presence of vector control, HA-NP, or HA-VP35. Representative images of (**A**) endogenous RUVBL1 localization pattern with control vector (top panels), HA-NP (middle panels), or HA-VP35 (bottom panels) and (**B**) endogenous RUVBL2 localization pattern with control vector (top panels), HA-NP (middle panels), or HA-VP35 (bottom panels) are shown. HA-NP or HA-VP35 (green), RUVBL1/2 (red), and Hoechst 33342 nuclear stain (blue) were visualized by confocal microscopy. Scale bars = 20 µM.

**Figure 4 viruses-11-00372-f004:**
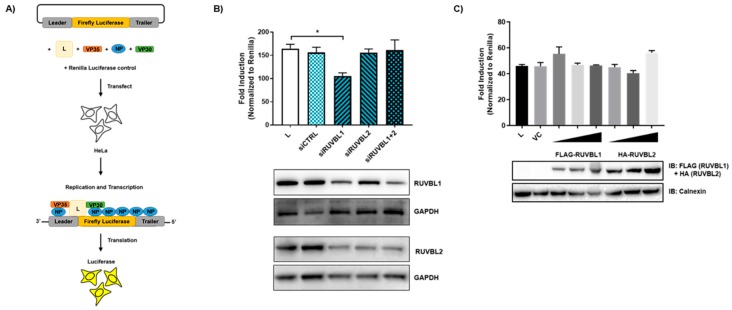
RUVBL1/2 do not effect EBOV minigenome activity. (**A**) Schematic diagram of the EBOV minigenome system. The EBOV minigenome system consists of six plasmids: Four support plasmids encode replication complex components NP, L, VP35, and VP30. The EBOV minigenome plasmid encodes a firefly luciferase reporter gene flanked by the leader and trailer sequences of EBOV, and the plasmid that encodes *Renilla* luciferase is used for normalization. (**B**) Minigenome activity upon the knockdown of either RUVBL1, RUVBL2, or in combination. Below are protein levels confirmed by immunoblot. HeLa cells were transfected with 80 nM scrambled siRNA, 30 nM siRNA targeting RUVBL1, or 50 nM siRNA targeting RUVBL2. Twenty-four h after siRNA addition, the minigenome components were transfected. Forty-eight h later, minigenome reporter activity was measured. (**C**) Overexpression of FLAG-RUVBL1 and HA-RUVBL2 in the EBOV minigenome. HeLa cells were left untransfected, or transfected with vector control (VC), or increasing amounts of FLAG-RUVBL1 (125, 250, and 500 ng) or HA-RUVBL2 (125, 250, and 500 ng). Twenty-four h after exogenous transfection, the minigenome components were transfected. Forty-eight h later, minigenome reporter activity was measured. Data represent mean ± SEM from one representative experiment (*n* = 3) of at least three experiments (* *p* < 0.05).

**Figure 5 viruses-11-00372-f005:**
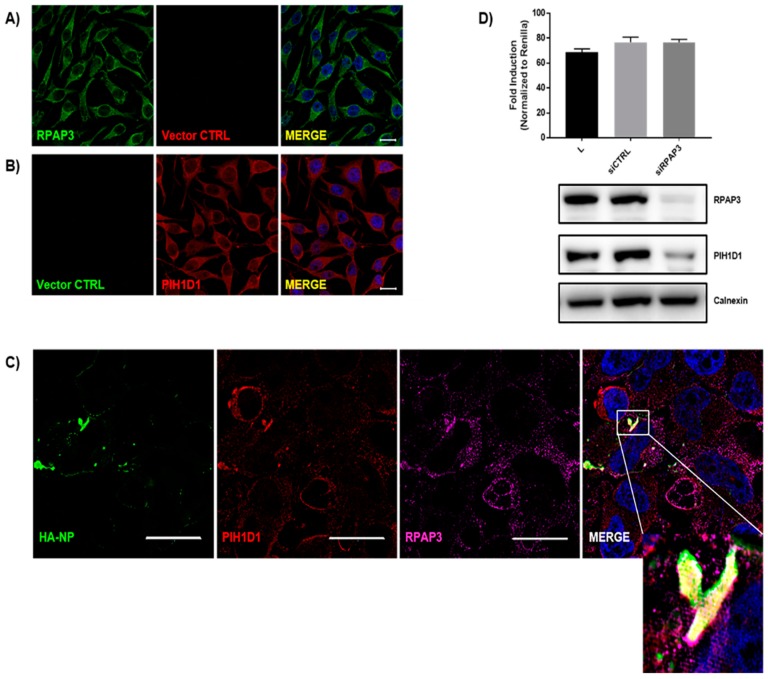
The R2TP complex components RPAP3 and PIH1D1 colocalize with HA-NP. HeLa cells were transfected with vector control or HA-NP. Twenty-four hours later, the cells were fixed and processed for the immunofluorescence detection of endogenous PIH1D1 or RPAP3 in the presence of vector control or HA-NP. Representative images of endogenous localization pattern of (**A**) RPAP3 with vector control and (**B**) PIH1D1 with vector control. RPAP3 or vector control (green), PIH1D1 or vector control (red), and Hoechst 33342 nuclear stain (blue) were visualized by confocal microscopy. Scale bars = 20 µM. (**C**) Representative images of endogenous localization pattern of PIH1D1 and RPAP3 in the presence of HA-NP. HA-NP (green), RPAP3 (red), PIH1D1 (magenta), and Hoechst 33342 nuclear stain (blue) were visualized by SIM. Scale bars = 20 µM. (**D**) Minigenome activity upon the knockdown of RPAP. Below are protein levels confirmed by immunoblot. HeLa cells were transfected with 30 nM scrambled siRNA or 30 nM siRNA targeting RPAP3. Twenty-four hours after siRNA addition, the minigenome components were transfected. Forty-eight hours later, minigenome reporter activity was measured. Data represent mean ± SEM from one representative experiment (*n* = 3) of at least three experiments.

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
