# Peer review of "Identification of RUVBL1 and RUVBL2 as Novel Cellular Interactors of the Ebola Virus Nucleoprotein"

_viruses, 2019, doi:10.3390/v11040372_

Round 1

Reviewer 1 Report

The manuscript by Marwitzer et al reports on the identification of two cellular proteins, RUVBL1 and RUVBL2, as proteins interacting with the nucleoprotein (NP) of Ebola virus.

The study is of particular relevance because the NP interactome, despite the viral protein high abundance in the infected cells, has been little characterized until now.

In my opinion the manuscript is of great interest and deserves publication. My major observations are:

In lines 190-192 authors mention that “Comparing these results to two previous studies that investigated NP cellular interactors revealed an overlap of seven cellular proteins that have a higher probability of forming protein-protein interactions, respectively [17,22].” They mention the other proteins in the discussion (lines 376-377) however, it could be useful to report them also in the results. In fact, it is not really clear to me the rationale based on which authors focused on only RUVBL1/2 proteins identified in reference 22 and not on the others (see lines 198-199).

While I have to say that I don't know much about RUVBL1 and RUVBL2, from what authors write seems to me that they interact one with the other. If this is the case, based on the presented results, I wonder whether there is a direct interaction between NP and both proteins or if NP mainly interacts mainly with RUVBL1 that then interacts with RUVBL2. It seems to me that authors suggest this possibility when they state in lines 268-269 “That only RUVBL1 knockdown is observed to affect minigenome activity suggests RUVBL1 may drive the interaction with NP”. If it is known that RUVBL1/2 work as a complex, authors should comment on this and, eventually, perform an IP in the presence of siRNA silencing of RUVBL1 and a RUVBL2 expression plasmid (since the knockdown of one influences the expression of the second) to check RUVBL2 direct interaction with NP. Alternatively, authors should check if in the HA-RUVBL2 IP they also find RUVBL1 and, vice versa, if in FLAG-RUVBL1 IP they find RUVBL2 also.

I wonder why in figure 5 authors showed only RPAP3 and PIH1D1 and NP colocalization and did not use also the anti- RUVBL1 and RUVBL2 antibodies. Do authors think this to be redundant? Could authors add these data?

Author Response

See the attached document please.

Reviewer 2 Report

Comments to Authors:

The manuscript by Morwitzer et al. presents the identification of host proteins that can interact with the Ebola virus (EBOV) nucleoprotein.  Through mass spectrometry they identify RUVBL1 and RUVBL2 as candidate NP interactors and visualize the interactions of these proteins through reciprocal immunoprecipitations and immunofluorescence mincroscopy.  They evaluate the effect of the absence of RUVBL1 on EBOV mini genome transcription and visualize the co-localization of RUVBL1-interactors with overexpressed EBOV NP.

The motivation of this manuscript is to identify novel host proteins that interact with NP, and with this knowledge identify novel therapeutic targets for EBOV.

I agree with the authors – virus/host interactions are a fascinating field and host-directed therapeutics is an attractive arena for further exploration.  Additionally, NP is an important component of EBOV replication and inhibiting its machinery could have a large impact on viral replication.

Overall, I enjoyed reading this manuscript – it is well-written and the authors have made a good first attempt.  However, their observations would be significantly strengthened if they included some additional controls.  Unfortunately, in its current state the manuscript is not acceptable to publication in Viruses.  However, if the authors can provide the additional data, and provide some requested clarification, then I think that it can be re-considered for publication. 

Major Concerns:

A major challenge during overexpression experiments is that one needs to demonstrate that overexpression mimics endogenous protein-protein interactions, and one needs to demonstrate that the protein tags are not responsible for the observed protein-protein interactions. Comments related to these challenges:

-The HA vector was used as a control for the first figure, but is not used as a negative control in subsequent figures - 3 and 5 (microscopy images) (I’m not sure if it was used as a control in figure 2?).  This data will strengthen the authors’ arguments that the RUVBL1 protein and RUVBL1/2 interactors are specifically binding to NP and not the HA tag.  It will also demonstrate that the HA signal in green is not bleeding over into the other wavelengths (and vice versa). 

            -Some AAA+ proteins are involved in disaggregating protein aggregates.  It is hard to tell if the RUVBL1/2 interactions are genuine and not a consequence of NP overexpression.  Including HA controls would better support the authors’ claim that the RUVBL1/2 proteins are true NP-specific interactions.

-Overexpression does not always recapitulate a live viral infection, as the authors acknowledge in their discussion.  Have the authors attempted to visualize RUVBL1/2 localization during a live viral infection?  (Asking because co-authors from USAMRIID and Sina Bavari’s lab are included on this manuscript).  For someone with access to a BSL4 lab, fixed or gamma-irradiated slides could be easily prepared and microscopy staining could occur under BSL2 conditions, instead of BSL4 conditions.

-I appreciate that the authors identified an NP-host protein interaction and then extended their observations into a potential mechanism.  However, the results in Figure 4B (knockdown results) are curious.  I would have expected that knockdown of RUVBL1 and RUVBL2 would also reduce luciferease levels to a similar extent as knockdown of RUVBL1 alone.  But, this is not what the data present. 

One potential idea is that loss of RUVBL1 results in an excess of RUVBL2 free subunits and the free subunits of RUVBL2 interfere with MG activity (instead of RUVBL1 being necessary for MG activity).  Have the authors tried overexpressing RUVBL2 and evaluating its effect on MG activity?  I would hypothesize that RUVBL2 overexpression would also reduce MG activity.  The results in Figure 4B are a bit odd, but overexpressing RUVBL1/2/HA could yield some interesting insights.

-Did the authors try knocking down RPAP3, and/or PIH1D1, too?  This data would better tie the siRNA knockdown data (figure 4) to the microscopy data in the subsequent figure (figure 5).   

Minor Concerns:

-Perhaps I missed it, but I did not see a reference to the supplementary table. Also, can the authors include a spreadsheet with the proteins that were immunoprecipitated with the vector (negative) control?  The labeling in the supplementary figure is not very well explained.  What does NP-1 and NP-2 mean?

-The scale bars and scale bar text for Figure 3 are difficult to see and need to be bigger.

-Why did the authors transfect in different amount of siRNA (30nM vs 50nM), instead of a consistent amount?

-Did the authors try blotting for RPAP3 and PIH1D1 in their immunoprecipitation experiments?

Round 2

Reviewer 1 Report

In my opinion the answer of the authors to my comments are adequate and the paper can be accepted.

Author Response

We thank the reviewer for their comments.

Reviewer 2 Report

The authors have addressed my concerns with the additional data in their rebuttal letter and I suggest that they move the data in the rebuttal figures into the main figures in the manuscript..  My most significant concern was that they did not include enough controls to verify that the RUVBL1/2 interaction with NP was a true interaction and not the result of RUVBL1/2 interacting with the HA tag on NP.  I think that their manuscript is significantly strengthened with these additional controls.  Specific requests are below:

-Can the authors add rebuttal Figure 2 to Figure 3?  While they do not include a staining control for the HA tag in Figure 3 (no green channel was visualized with the vector control), the data in Rebuttal figure 2 clearly demonstrates that the RUVBL1/2 interaction with NP is specific and not the consequence of protein overexpression, nor a result of interactions with the HA tag.  This is some very nice data and should be included in the figure.

-Can the authors add the data from rebuttal figure 1 to Figure 5 in the manuscript?  These controls are missing from Figure 5 and clearly demonstrate that PIH1D1 and RPAP3 localization is specific to HA-NP.

- Initially, it was not clear that Morwitzer et al. included a vector control with each of their experiments, however, the rebuttal letter provides additional clarification.  This vector control data is somewhat buried in the materials and methods.  Can the authors please update the figure legends for Figures 3 and 5 to clearly state that the correct controls were included?

-I appreciate the extra MG expression data that was included in the rebuttal letter (rebuttal figures 3 and 4).  The authors’ comment, “So, while the decrease in MG activity with RUVBL1 knockdown is statistically significant, at this time we do not believe it to be functionally relevant”  is especially insightful, given the extra data that they include in the rebuttal letter.  Can the authors add the rebuttal figure 3 and 4 data to Figure 4?  I, like the authors, am usually hesitant to include negative data in a manuscript, but I think that this negative data will help set the stage for their hypothesis that RUVBL1/2 are involved in capsid assembly.

Author Response

Please see attached comments.
